# Trabectedin Is Active against Two Novel, Patient-Derived Solitary Fibrous Pleural Tumor Cell Lines and Synergizes with Ponatinib

**DOI:** 10.3390/cancers14225602

**Published:** 2022-11-15

**Authors:** Bahil Ghanim, Dina Baier, Christine Pirker, Leonhard Müllauer, Katharina Sinn, Gyoergy Lang, Konrad Hoetzenecker, Walter Berger

**Affiliations:** 1Department of Thoracic Surgery, Medical University of Vienna, 1090 Vienna, Austria; 2Center for Cancer Research and Comprehensive Cancer Center, Medical University of Vienna, 1090 Vienna, Austria; 3Department of General and Thoracic Surgery, Karl Landsteiner University of Health Sciences, University Hospital Krems, 3500 Krems, Austria; 4Institute of Inorganic Chemistry, University of Vienna, 1090 Vienna, Austria; 5Research Cluster “Translational Cancer Therapy Research”, 1090 Vienna, Austria; 6Department of Pathology, Medical University of Vienna, 1090 Vienna, Austria

**Keywords:** solitary fibrous tumor, *NAB2-STAT6* gene fusion, targeted therapy, trabectedin, ponatinib, dasatinib, in vitro, patient-derived cell lines

## Abstract

**Simple Summary:**

Solitary fibrous tumor of the pleura (SFT) is an orphan disease resistant to standard systemic therapy. We managed to establish two patient-derived cell models characterized as SFT by the *NAB2-STAT6* gene fusion. Cell lines were tested for drug responsiveness in vitro. Trabectedin and distinct multi-tyrosine kinase inhibitors were effective as single agents. Most interestingly, the combination of trabectedin with ponatinib or dasatinib showed synergistic effects against fusion-positive SFT cell viability, thus suggesting two novel, potentially interesting treatment regimens for this rare and, to date, treatment-refractory disease.

**Abstract:**

Solitary fibrous tumor of the pleura (SFT) is a rare disease. Besides surgery combined with radiotherapy in nondisseminated stages, curative options are currently absent. Out of fourteen primo-cell cultures, established from surgical SFT specimens, two showed stable in vitro growth. Both cell models harbored the characteristic *NAB2-STAT6* fusion and were further investigated by different preclinical methods assessing cell viability, clone formation, and protein regulation upon single-drug treatment or in response to selected treatment combinations. Both fusion-positive cell models showed—in line with the clinical experience and the literature—a low to moderate response to most of the tested cytotoxic and targeted agents. However, the multi-tyrosine kinase inhibitors ponatinib and dasatinib, as well as the anti-sarcoma compound trabectedin, revealed promising activity against SFT growth. Furthermore, both cell models spontaneously presented strong FGFR downstream signaling targetable by ponatinib. Most interestingly, the combination of either ponatinib or dasatinib with trabectedin showed synergistic effects. In conclusion, this study identified novel trabectedin-based treatment combinations with clinically approved tyrosine kinase inhibitors, using two newly established *NAB2-STAT6* fusion-positive cell models. These findings can be the basis for anti-SFT drug repurposing approaches in this rare and therapy-refractory disease.

## 1. Introduction

Solitary fibrous tumor of the pleura (SFT) belongs to the group of soft tissue tumors and can be regarded as an orphan disease with an annual incidence of less than 0.1 per 100,000 in Europe [1]. SFT is defined as mesenchymal neoplasm and was previously divided into a benign and malignant subtype by the England criteria [2]. In 2002, de Perrot et al. suggested a staging system including macroscopic and histologic growth patterns [3]. Both the tumor dignity and the de Perrot staging proved to estimate the probability of recurrence [4]. Nevertheless, the most recent, fifth edition of the WHO classification suggests to avoid the terms benign and malignant and instead use risk classification models since the previous histological stratification did not accurately reflect the clinical behavior [5]. Regarding the genetic characterization, the *NAB2-STAT6* gene fusion was described as a distinct hallmark of SFT. The fusion protein promotes the malignant phenotype via constitutive activation of early growth response 1 (EGR1)-mediated gene transcription by inhibiting the EGR1-repressing activity of wild-type NAB2 [6,7]. In addition to the fusion transcript detection, nuclear STAT6 staining can be useful in the diagnosis of SFT besides the traditional immunohistochemical markers vimentin, CD34, CD99, and bcl-2 [4,8]. Moreover, fibroblast growth factor receptor 1 (FGFR1) and its ligand fibroblast growth factor 2 (FGF2) are overexpressed in SFT [9,10], the ligand being even suggested as a prognostic parameter [11].

The clinical outcome after resection with a curative intention for SFT is—compared to other thoracic malignancies—excellent, with 77% and 67% of patients being still alive without evidence of disease after 5 and even 10 years, respectively, as demonstrated before by our study group in a large international multicenter study analyzing 125 pleural SFT patients [12]. Today, radical surgery is the only treatment modality able to cure the disease and chemo- or radiotherapy alone failed to distinctly improve clinical outcome, resulting in a significant lack of noninvasive treatment alternatives for patients presenting with unresectable SFT, i.e., due to enhanced tumor size or dissemination [13]. In addition, even completely resected tumors can recur decades after radical resection and not all patients are eligible candidates for thoracic surgery or redo surgery at the time of later recurrence. Thus, improving the systemic treatment of SFT, either as part of multimodality therapy concepts including surgery or used alone to treat the unresectable disease, is urgently warranted [13,14,15].

With regard to systemic therapy, most studies were—according to the low incidence of SFT—of a retrospective nature and analyzed only small numbers of patients [16]. In these studies, low to moderate response rates were reported for conventional chemo-, as well as targeted therapy [14,15,17,18,19]. Based on this knowledge, we aimed to establish patient-derived SFT cell models to test modern anticancer compounds and their combinations in vitro in frame of a drug repurposing approach for this otherwise treatment-resistant orphan disease. To acknowledge the most recent major breakthrough with regard to the *NAB2-STAT6* fusion characterizing SFT as a translocation-related soft tissue tumor, we focused on therapy approaches that proved to previously be efficient in translocation- and fusion-related malignant diseases [5].

## 2. Materials and Methods

### 2.1. Establishment of Patient-Derived Cell Lines

All patients with a clinical SFT diagnosis receiving tumor resection between 12/2014 and 12/2021 at the Medical University of Vienna, Department of Thoracic Surgery were included. Tumor samples were retrieved during surgery and used for cell culturing after written consent was granted by each included patient. The study was approved by the Ethics Committee of the Medical University of Vienna under the title, “Etablierung einer Tumorbank für die molekulare Analyse von thorakalen Tumoren”, EK NR.: 9004/2009. In brief, a small part of the tumor was removed in the operating room for tumor cell isolation and brought to the Center for Cancer Research and Comprehensive Cancer Center, Medical University of Vienna. The tumor probe was shredded and then grown in ACL-4 medium (ATCC MEDIUM 8002) supplemented with human epidermal growth factor (1 ng/mL, E9644), hydrocortisone (10 µg/mL, T2036), transferrin (24.2 ng/mL, H2270), and fetal calf serum (10%). Penicillin (100 U/mL) and streptomycin (10 mg/mL), all obtained from Sigma-Aldrich (St. Louis, MO, USA), were added to the medium for primary culture establishment. The cells were cultured in a humidified incubator at 37 °C and 5% CO_2_. After demonstrating stable in vitro growth, the cell lines were further investigated to prove their SFT origin, as later described in the Results section.

### 2.2. Immunohistochemistry

Routine histopathological processing, including immunohistochemistry (IHC), of the patients’ tumor tissues was performed at the Department of Pathology of the Medical University of Vienna. The antibody panels used for immunohistochemistry included bcl-2 (clone 124; DAKO Agilent, Santa Clara, CA, USA), CD34 (clone QBend/10; Leica, Wetzlar, Germany), CD99 (clone EP8, Biocare, Pacheco, CA, USA), STAT6 (anti-Stat 6 polyclonal antibody; Sigma-Aldrich, St. Louis, MO, USA), and Ki67 (clone 30-9; Ventana, Oro Valley, AZ, USA). In addition, all cases were re-reviewed by the author LM to confirm pathologic SFT diagnosis.

### 2.3. Detection of the NAB2-STAT6 Fusion

RNA was isolated from formalin-fixed and paraffin-embedded tumor samples (FFPE) and the corresponding cell lines. The *NAB2-STAT6* fusion was analyzed by using next-generation sequencing (NGS). For both materials (FFPE tissue and corresponding cell lines), the TruSight RNA Fusion Panel (Illumina, San Diego, CA, USA) was used for library generation. Sequencing was performed with a MiSeq instrument (Illumina, San Diego, CA, USA). All NGS results were verified by reverse transcription PCR as described below in Section 2.9.

### 2.4. Drugs

Ponatinib was purchased from LC Laboratories (Woburn, MA, USA). Trabectedin was obtained from PharmaMar (Colmenar Viejo, Spain). Cisplatin was kindly provided by the Institute of Inorganic Chemistry, University of Vienna. Ponatinib, dasatinib, obatoclax mesylate, venetoclax, vincristine, PD173074, and stattic were obtained from Selleckchem (EUBIO, ANDREAS KÖCK e.U., Vienna, Austria). Nintedanib and imatinib were purchased from LC Laboratories (Woburn, MA, USA). Doxorubicin was acquired from Sigma-Aldrich (St. Louis, MO, USA). Paclitaxel was procured from Bristol Myers Squibb (New York City, NY, USA).

### 2.5. Determination of Cell Proliferation

For the determination of cell proliferation, SFT-T1, SFT-T2, HCC827, and Met5a cells were seeded at a density of 3.5 × 10^3^ cells per well in triplicates in 96-well plates in 300 µL of ACL medium or RPMI-1640 (Sigma-Aldrich, St. Louis, MO, USA) supplemented with 10% fetal calf serum and incubated overnight. Cells were imaged the next day (T0), and 24, 48, and 72 h later using the Cytation 5 Cell Imaging Multimode Reader (BioTek, as part of Agilent, Winooski, VT, USA). A digital phase contrast was created from the derived images. Following this, pixel intensities per well were quantified by using ImageJ 1.50i (NIH, Bethesda, MD, USA).

### 2.6. Cell Viability Assay

The 3-(4,5-dimethylthiazol-2-yl)-2,5-diphenyltetrazolium bromide (MTT) assay (EZ4U, Biomedica, Vienna, Austria) was utilized as a cell viability and drug sensitivity assay to screen for the in vitro responsiveness of our cell lines. SFT cell lines were seeded at a density of 3.5 × 10^3^ cells/100 µL in 96-well microtiter plates and incubated overnight. Cells were exposed to the respective single drugs and drug combinations for 72 h. Cell viability was measured as published before [20]. From full dose–response curves, the respective IC_50_ values were calculated by nonlinear regression curve-fitting (sigmoidal dose–response with variable slope). To evaluate the efficacy of drug combinations, the combination index (CI) was calculated according to the method published by Chou and Talalay [21] using the CalcuSyn software (Biosoft, Ferguson, MO, USA).

### 2.7. Clonogenicity (Clone Formation) Assay

For clonogenic assays, cells were seeded at a density of 1000 cells per well in 500 µL of medium in 24-well microtiter plates and allowed to adhere overnight. Cells were exposed to the indicated single drugs and combinations for nine days. After washing with phosphate-buffered saline (PBS, pH 7.4; Sigma-Aldrich Inc., St. Louis, MO, USA), cells were fixed with acetone, and stained with 0.01% (*w/v*) crystal violet. Crystal violet was re-solubilized with 2% sodiumdodecylsulfate (Sigma-Aldrich, St. Louis, MO, USA) and the fluorescence intensity was measured on a spectrophotometer (Tecan Infinite 200 Pro, Tecan Trading AG, Männedorf, Switzerland). 

### 2.8. Western Blot Analyses

For Western blot analyses, cells were seeded into 6-well plates at a density of 4 × 10^5^ cells per well in 2 mL of medium and incubated overnight. Cells were exposed to the respective drugs and combinations for the indicated time, and thereafter, proteins were harvested and analyzed by gel electrophoresis and immunoblotting as published before [20]. FGF2 was acquired from PeproTech (FGF2, Cranbury, NJ, USA). For FGF2 stimulation, cells were starved (medium without FBS) for 24 h prior to treatment. Cells were treated with the indicated drug concentrations for 1 h. Then, 15 min before the end of the treatment time, FGF2 was added for 15 min at a concentration of 20 ng/mL. Sample collection, protein isolation, separation, and Western blotting were performed as described previously [22]. Primary antibodies phospho-p44/42 MAPK (p-Erk) (Thr202/Tyr204) (20G11) (#4376, dilution 1:1000), Erk1/2 (#4695, 1:1000), p-Akt (Ser473) (D9E) (#4060, 1:500), Akt (pan) (C67E7) (#4691, 1:1000), p-Src (Tyr416) (#6943, 1:500), Src (36D10) (#2109, 1:1000), p-S6 ribosomal protein (Ser240/244) (D68F8) (#5364, 1:1000), STAT6 (D3H4) (#5397, 1:1000), and vimentin (R28) (#3932, 1:1000) were purchased from Cell Signaling Technology (Danvers, MA, USA). Anti-Flg (FGFR1; C-15) (sc-121, 1:250) and S6 (C-8) (sc-74459, 1:1000) were acquired from Santa Cruz Biotechnology (Dallas, TX, USA). Anti-ß-actin (AC-15) (A5441, 1:2000) was obtained from Sigma-Aldrich (St. Louis, MO, USA). Horseradish peroxidase-linked secondary antibodies, anti-mouse IgG (Fc specific) antibody (A0168) and anti-rabbit IgG antibody (7074S) were purchased from Merck KGaA (Darmstadt, Germany) and Cell Signaling Technology (Danvers, MA, USA). 

### 2.9. RNA Isolation, Reverse Transcription into cDNA, and RT-PCR

Total RNA was isolated with Trizol using standard protocols and reverse transcribed into cDNA as described in Berger et al. [23]. An RT-PCR was performed to screen for the presence of the *NAB2-STAT6* fusion gene using the primer sequences (Set 1) and the PCR conditions described by Guseva et al. [24]. *GAPDH* served as a housekeeping gene (*GAPDH fw:* 5′-CTG GCG TCT TCA CCA CCA T-3′; *GAPDH rev:* 5′-GCC TGC TTC ACC ACC TTC T-3′).

### 2.10. Statistical Analyses

Metric data are given as mean ± SD if not indicated otherwise. An unpaired Student’s *t*-test was used to detect significant differences between two groups. A one-way analysis of variance (ANOVA) was used to compare means of more than two groups, and the Bonferroni or Tukey post-test were utilized to correct for multiple testing. *p*-values below 0.05 were considered statistically significant: * *p* < 0.05, ** *p* < 0.005, *** *p* < 0.0005, and **** *p* < 0.0001. All calculations and graphs were performed with Prism 8.0 software (GraphPad, San Diego, CA, USA).

## 3. Results

### 3.1. Establishment of Novel SFT Models from Surgical Specimens

In brief, fourteen primo-cell cultures were derived from patients with clinical SFT suspicion during their surgical tumor resection between 2014 and 2021. To prove that all included 14 cell lines were of SFT origin, we first reviewed postoperative pathology reports of the respective patients. Four cell lines had to be excluded because the final histology did not confirm clinical SFT suspicion. Out of the remaining ten cell lines, five turned out to be senescent or did not grow stably in vitro. For the final five stable-growing cell lines, the corresponding patient tumor tissue of the pathology archive was re-evaluated and tested for the *NAB2-STAT6* fusion by NGS to confirm the clinical and histological SFT diagnosis on a genetic level. In three of these cell lines, the genomic fusion event was missing, despite its presence in the original surgical specimen. Thus, finally, two patient-derived SFT cell models stably expressing the *NAB2-STAT6* fusion protein were established (Figure 1), reflecting, on the one hand, the rarity of the disease, and on the other hand, the challenge to culture SFT in vitro. One cell line was derived from the low-risk group SFT (SFT-T1) and the other from the intermediate-risk group SFT (SFT-T2), according to the Demicco classification [25], as summarized in Table 1.

Representative pictures of the surgical tumor specimens from SFT-T1 and SFT-T2 showing the general microscopic growth pattern and the IHC profile are given in Figure 2. Both tumors were (immune-)histologically fitting well to the pathological diagnosis of SFT (Table 1). Tumor cells were positive with regard to the membranous and cytoplasmic expression of CD34, bcl-2, and CD99 and furthermore showed nuclear STAT6 positivity (Figure 2b,c,d, and f, respectively). Fibromatosis (Fibr) served as a pathological negative control, and STAT6 and CD34 staining was absent in the respective tissue sample. Furthermore, negative control staining with isotypes of the corresponding antibodies can be found in Appendix A. Expression of proliferation marker Ki67 (Figure 2e) is lower in the low-risk group representative SFT-T1 and fibromatosis tissue as compared to that in intermediate-risk SFT-T2. The corresponding quantification of Ki67 is given in Table 1.

Both cell lines demonstrated stable in vitro proliferation (Appendix A) and a mesenchymal growth pattern compared to non-small-cell lung cancer HCC827 and pleural mesothelial Met5a cells with epithelial morphology, as shown in Figure 3a. Furthermore, SFT marker expression was validated by Western blot analyses in vitro. Expression of the SFT marker proteins bcl-2 and vimentin (Figure 3b, original Western blots Appendix A) in the cell models reflected the immunohistochemical results of the original patient material (compare Figure 2) and the pathological SFT diagnosis as shown in Table 1. In addition, the presence of the fusion oncogene detected by NGS was verified on the RNA level by reverse transcription PCR (Figure 3c). Fitting to the predicted size of the *NAB2-STAT6* fusion protein, an aberrant Western blot STAT6 signal was detected at 135 kDa in SFT-T1 and SFT-T2 as compared to the wild-type STAT6 signal at 110 kDa expressed by the immortalized pleura cell line Met5a and non-small cell lung cancer HCC827 cells (Figure 3d, original Western blots Appendix A).

Interestingly, NGS analysis of both cell lines and the tumor tissue from all five patients exhibited the identical *NAB2-STAT6* fusion (NAB2 chr12:57486749:+ STAT6 chr12:57502081:-; NAB2 exon 4/STAT6 exon 2; HG19), as demonstrated in Figure 4. As previously published, this variant of the *NAB2-STAT6* fusion is the most prevalent alteration in pleural SFT [26].

### 3.2. Screening for Treatment Responses in Novel SFT Cell Models

After establishing the two SFT cell models with a confirmed *NAB2-STAT6* fusion, we first screened for the efficacy of different chemotherapeutic and targeted compounds to obtain a general idea of SFT-specific in vitro drug responsiveness. Representative drug–response curves are provided in Appendix A. The response to the widely used anthracycline doxorubicin was limited in both, the cell line derived from the low-risk SFT (SFT-T1) and the one derived from the intermediate-risk SFT (SFT-T2). This is in line with the literature, where the response to doxorubicin was limited to the dedifferentiated subtype, as reviewed before [16]. Similar results were found for other commonly used cytotoxic compounds including cisplatin and taxol, as summarized in Table 2. 

From the tested drugs reported thus far, DNA-interfering trabectedin and the multi-tyrosine kinase inhibitors (TKIs) ponatinib, dasatinib, and nintedanib, as well as the more FGFR1-specific PD173074, showed pronounced activity against SFT tumor growth (Table 2). Furthermore, the bcl-2 inhibitor obatoclax was moderately active when compared to, e.g., colorectal cancer cell lines [27] or bladder cancer [28]. Of note, the highly efficient compounds ponatinib and dasatinib, as well as trabectedin, are well known to be active against translocation-/fusion-related malignant diseases as summarized before [29,30], and thus might also be suitable therapies against fusion-positive SFT. 

### 3.3. Ponatinib and Trabectedin Are Active against SFT Cell Growth and Synergize In Vitro

Trabectedin demonstrated high anti-SFT activity in both tested cell models as a single-agent treatment with an IC_50_ in the low nM range. Ponatinib was also highly effective as a single drug with low IC_50_ values in the high nM to low µM range. Respective dose–response curves are given in Figure 5a,b. Upon combination of ponatinib and trabectedin, a strong synergistic effect of this regimen was observed in cell viability assays with CI values < 0.9 (Figure 5c,d).

The synergistic effect of trabectedin and ponatinib was, furthermore, tested in clone formation assays for a longer treatment duration (Figure 5e,f). In SFT-T2 cells, clone formation was significantly reduced by ponatinib and trabectedin as single compounds. Upon combination of the two drugs, the clone formation capacity was further inhibited at low doses of both drugs, verifying the observations from short-term cell viability testing. In SFT-T1 cells, comparable results were obtained (Appendix A).

### 3.4. Ponatinib Treatment Targets Fibroblast Growth Factor Receptor Downstream Signaling in SFT Cells

For an in-depth evaluation of the mode of action of ponatinib in SFT-T1 and SFT-T2 cells, Western blot analyses were performed (Figure 6, original Western blots Appendix A). The expression of the FGFR downstream signaling proteins of the MAPK and PI3K/Akt pathway [31], as well as their activating phosphorylation, was analyzed after exposure to ponatinib [30]. Since the *NAB2-STAT6* fusion is linked to the activation of the FGFR signaling cascade [5], we also analyzed the effects of FGF2 stimulation on its downstream targets in SFT with and without FGFR inhibition. Of note, SFT responded strongly to FGF2 exposure by activating both the PI3K/Akt and MAPK pathway, indicated by Akt and Erk phosphorylation, respectively. This suggests an important role of FGF signaling in SFT as hypothesized [5] and demonstrated in SFT patients before [11]. Importantly, ponatinib efficiently inhibited both basal and FGF2-induced FGFR downstream signaling pathways, represented by the loss of phosphorylation of Erk and Akt. The complete blockade of the PI3K/Akt pathway was confirmed at the level of the downstream mediator S6. Of note, SFT-T1 cells exhibited phosphorylation of S6 without FGF2 stimulation in contrast to SFT-T2, which showed no basal S6 phosphorylation and less basal Akt phosphorylation. Additionally, basal FGFR1 expression was stronger in SFT-T2 compared to SFT-T1 cells. Surprisingly, the phosphorylation of Src (also, besides FGF signaling, a ponatinib target as reviewed before in [30]) was unaffected by the FGFR inhibitor in our fusion-positive SFT cells. In addition, Src expression was slightly enhanced after ponatinib exposure, indicating a potential dose- or specificity-dependent impact on the Src kinase as suggested before [32]. 

### 3.5. Dasatinib Is Active against SFT Growth In Vitro and Enhances Trabectedin Activity

Similar to ponatinib, dasatinib showed significant preclinical activity against our two fusion-positive cell models in the nM range (compare Table 2). A representative dose–response curve is depicted in Figure 7a. However, the first clinical data about dasatinib activity against sarcoma growth that included 25 SFT patients were disappointing [33]. Nevertheless, our observations suggest that dasatinib is active against SFT in vitro. Furthermore, the combination of dasatinib with trabectedin proved additive to synergistic efficacy in cell viability assays as demonstrated by CI-values below 1.2 or 0.9, respectively (Figure 7b,c)). In line with these findings, the dasatinib/trabectedin combination also further decreased clone formation as compared to the single treatments alone over a longer treatment duration (Figure 7d), suggesting that the response of SFT toward dasatinib can be enhanced by trabectedin. This finding in the SFT-T2 models corresponded well with observations in the SFT-T1 cells at a higher trabectedin concentration (Appendix A).

## 4. Discussion

SFT is a rare disease accounting for less than 5% of all pleural tumors [34]. Concerning prognostic biomarkers, even the most validated SFT prognosticators such as the Demicco classification [25] failed to estimate the outcome in SFT of the bone [35], and thus, an established staging and risk stratification is still missing for this rare malignancy. The standard therapy for local SFT presents radical surgery with or without radiotherapy [12,16,36]. At the disseminated stage, curative systemic treatment options are currently not available. Hence, SFT deserves attention regarding in-depth investigations of potential treatment approaches including drug repurposing strategies with the aim to improve the prognosis for this progressive disease. However, most of the explored systemic treatment approaches failed to significantly improve outcomes of SFT patients thus far [13,14,15,16,33,37]. Due to the low incidence numbers, most of the studies on SFT were of a retrospective nature and/or included only small numbers of SFT patients. Furthermore, the availability of data derived from primary SFT cell and xenograft models is even scarcer [16]. Thus, in the present study we investigated different innovative treatment approaches in two newly established fusion-positive, SFT-derived cell models to deliver preclinical data for potential further validation in the clinical setting.

Despite the low incidence numbers and other challenges of successful SFT in vitro cultivation, we were able to establish—from 14 surgical specimens with an initial SFT diagnosis—two permanent SFT cell models. Of note, the cell models presented here are—to the best of our knowledge—the first SFT patient-derived cell lines harboring the *NAB2-STAT6* fusion, proven at the mRNA and protein level. Regarding preclinical SFT models, the *NAB2-STAT6* fusion positivity was, thus far, only reported in case of two SFT patient-derived xenograft (PDX) models, out of which only one exhibited nuclear localization of STAT6 [37]. It should be mentioned here that, in our cell models, we always detected the expression of wild-type STAT6 together with the fusion protein, as also described by Robinson et al. [7], suggesting a potential cooperative function of the fusion with the wild-type proteins. However, at least two additional permanent SFT cell cultures from our collection had lost the transgene expression during in vitro propagation despite the fact that the origin from the *NAB2-STAT6* RNA-positive patient sample was proven by Short Tandem Repeat analysis. Interestingly, the loss of the fusion protein was described as a marker of so-called “dedifferentiated SFT” [38]. Whether comparable processes happened during the in vitro propagation of our *NAB2-STAT6*-negative cell models or cell clones with fusion protein loss were already present in the original tumor is a matter of ongoing investigations.

In the current study, we focused on systemic treatment options for the classical fusion-positive genotype of SFT. Hence, we employed our two fusion protein-expressing, novel SFT models to test an extended panel of chemo- and targeted therapeutics concerning their anti-SFT activity. Moreover, we performed molecular analyses for the most promising compounds regarding the interaction with FGFR signaling, due to the previously reported overexpression of FGFR1 in SFT [9,10]. Another important aspect of this work included the evaluation of the most effective therapy combinations in the preclinical setting. Furthermore, the already approved compound trabectedin for sarcoma patients was used in both investigated combination regimens. This aspect might support clinical testing of our drug combinations.

In line with the clinical behavior of SFT, our fusion-positive cell models were insensitive to most of the tested systemic treatments, as shown in Table 2. Among the relatively active compounds was trabectedin with IC_50_ values in the low nM range. Trabectedin is a marine alkaloid with DNA-interacting and immune-stimulatory activities [39] and was approved for treatment of advanced soft tissue sarcoma and recurrent ovarian carcinoma [40]. Trabectedin showed remarkable activity in translocation-related sarcomas by modulating the corresponding fusion oncogenes, as reviewed before [29,41]. As a member of the translocation-related soft tissue sarcomas, early retrospective studies suggested the efficacy of trabectedin also against SFT [19,42,43]. Indeed, promising results of trabectedin in treating the first two available preclinical SFT PDX models have been reported [37]. The strong cytotoxic potential of trabectedin against our two fusion protein-positive SFT models further suggests a clinical evaluation of the compound against progressing SFT. Moreover, it was shown that trabectedin modulates the tumor microenvironment by targeting tumor-promoting inflammatory cell compartments [29]. This, indeed, might enhance the clinical efficacy of trabectedin, since tumor-promoting inflammation is known to play an important role in malignant pleural disease including SFT, as also demonstrated by our group [12,44].

Despite the distinct activity of trabectedin in the PDX models, the outcome was not curative and tumors relapsed [37]. Consequently, we screened for promising combination approaches with clinically approved anticancer agents to enhance the—in part—promising clinical and preclinical activity of trabectedin. Other highly active compounds identified in our drug screen were ponatinib (a clinically approved multi-kinase inhibitor of FGFR, PDGFR, VEGFR, Abl kinase, and c-Kit), and dasatinib (inhibitor of Bcr-Abl, Src, and c-Kit). Both of these compounds were shown to exert antiangiogenic properties, another reported vulnerability of SFT [16]. Ponatinib seemed to be especially interesting, since one of its central targets are FGFRs, including FGFR1, which is frequently overexpressed in SFT [9,10]. Additionally, the FGFR inhibitor regorafenib displayed the highest growth inhibitor effects out of all tyrosine kinase inhibitors tested against a dedifferentiated SFT PDX model [45]. In addition, the multi-tyrosine kinase inhibitor pazopanib (VEGFRs, PDGFR, and cKit), that also has some modest efficacy against FGF signaling, was active as first-line therapy in metastatic SFT in a prospective study [46]. However, when it comes to the dedifferentiated subtype in the aforementioned PDX model, the more FGF-targeting therapy regorafinib proved to have the highest anti-SFT activity, and pazopanib was only moderately active [45].

Thus, one might assume ponatinib as the most promising systemic treatment option for SFT among the multi-tyrosine kinase inhibitors due to its (1) antiangiogenic, (2) fusion/translation, and (3) FGFR-targeting characteristics [5,16,30]. The latter was identified in our in vitro analyses to play a crucial role. Indeed, our novel SFT models were also highly responsive to FGF2-mediated signals by upregulating MAPK as well as PI3K/Akt FGFR downstream pathways. Interestingly, not only FGF2-induced, but also the basal activation of these pro-oncogenic signaling cascades was clearly downregulated in SFT-T1 and even completely blocked in SFT-T2 by exposure to ponatinib. As the used SFT cell culture medium was devoid of any FGF supplementation, these observations strongly suggest an autocrine FGFR-based growth and survival signaling loop by endogenously expressed ligands playing a central role in basal MAPK signal maintenance in SFT. This assumption is supported by early studies suggesting FGF2 expression as a diagnostic and prognostic marker in SFT [11]. Of note, it has to be mentioned that three effective drugs out of the drug efficacy screening were also targeting the FGF axis (ponatinib, nintedanib, PD173074—compare Table 2), again underlining a potential interesting and targetable role of FGFR in SFT. 

Src as a target of ponatinib appears to be of minor importance in our SFT models since the expression and phosphorylation of Src were not changed by treatment with a multi-tyrosine kinase inhibitor. However, to further test the role of Src as a potential target in SFT, we used the Src, c-Kit, and Abl kinase inhibitor dasatinib. Dasatinib exerted substantial anti-SFT cell activity in our novel models. The mechanisms underlying the impact are enigmatic as neither c-Kit, Abl kinase, nor Src seem to be major players in the SFT malignant phenotype. Accordingly, the first clinical data about dasatinib activity against sarcoma growth including 25 SFT patients were disappointing [33]. Nevertheless, this poor clinical single-agent activity might be improved after combining dasatinib with trabectedin, a combination that was found to exert additive to synergistic anti-SFT growth activity in vitro in both novel SFT cell models.. These findings might also be an interesting issue for clinical follow-up studies. In general, it seems reasonable to combine trabectedin with another targeted therapy to achieve the best response rates in this treatment-resistant disease.

## 5. Conclusions

During the project, we were able to establish two patient-derived SFT cell lines. Both cell lines harbored the SFT-characteristic *NAB2-STAT6* fusion. This achievement facilitates in vitro drug testing in an orphan disease, where large, prospective, randomized, SFT-specific trials have been missing thus far. Our experiments provided evidence that fusion-positive SFT—as a generally treatment-resistant disease—is responsive to trabectedin in vitro. In addition, we were able to show that the tested SFT cell lines were strongly responsive to FGF2 stimulation, and basal MAPK/PI3K signaling was sensitive towards the FGFR inhibitor ponatinib. Furthermore, and of clinical relevance, we were able to propose synergistic treatment combinations, including trabectedin with the multi-tyrosine kinase inhibitors ponatinib and dasatinib. In particular, the combination of trabectedin with ponatinib might represent a promising treatment approach in this otherwise resistant disease, and is worth being validated in the clinical setting.

## Figures and Tables

**Figure 1 cancers-14-05602-f001:**
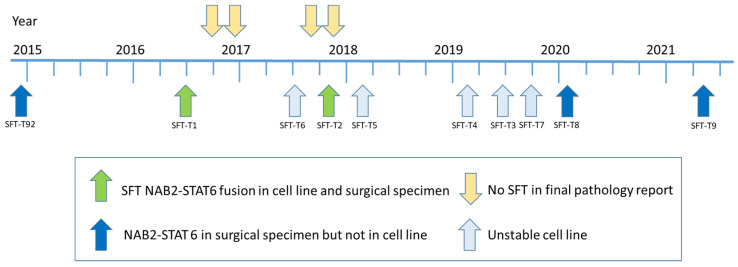
Timeline of study in- and exclusion. Out of 14 cell lines included during the seven-year project period, 2 were finally suitable to be used for the following experiments. In both, the patients’ tumor material as well as the cell lines, the NAB2-STAT6 fusion was detected and, accordingly, SFT origin was proven on the genetic and histologic level.

**Figure 2 cancers-14-05602-f002:**
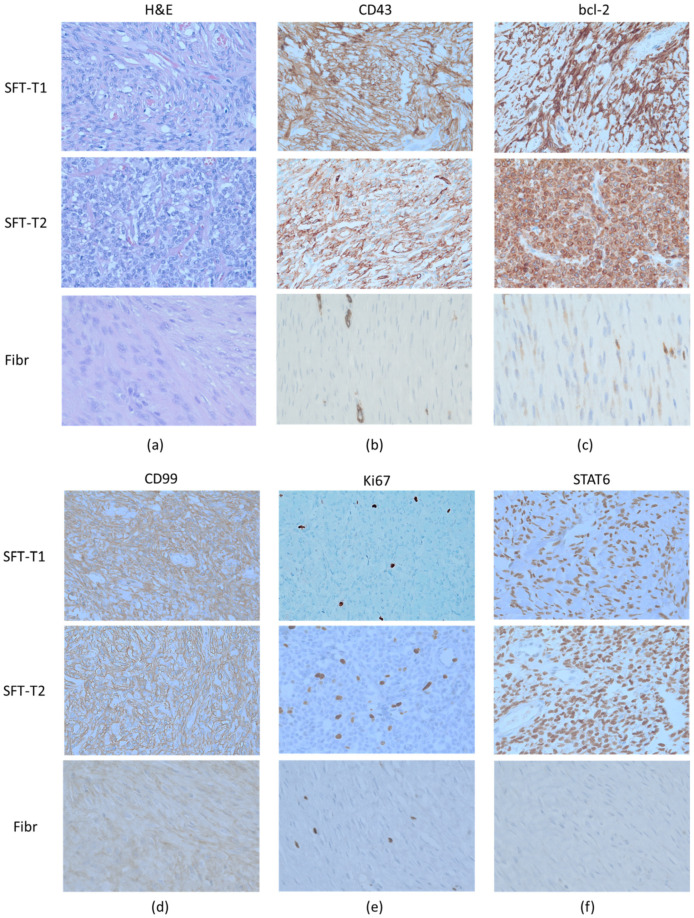
Representative (immune-)histological pictures of the two included tumors (SFT-T1 and SFT-T2) and an excluded fibromatosis patient (Fibr). The patient tumor samples showed the SFT typical pathomorphologic architecture in the H&E staining (**a**) and also the immunohistochemistry was positive for the diagnostic markers CD34 (**b**), bcl-2 (**c**), CD99 (**d**), and nuclear STAT6 staining (**f**), proving that both cell lines (SFT-T1 and -T2) were derived from SFT patients on the pathologic level. In addition, the low-risk group SFT-T1 showed less Ki67 (**e**) expression compared to its intermediate-risk group counterpart. In contrast to the SFT tumor tissue, fibromatosis was negative with regard to CD34 and STAT6 expression.

**Figure 3 cancers-14-05602-f003:**
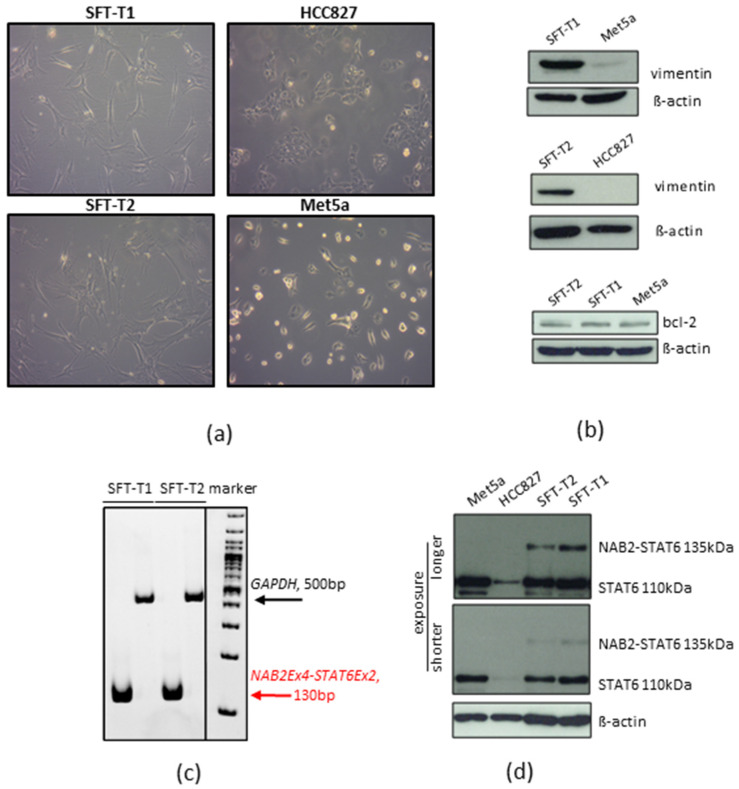
Growth and SFT marker protein expression pattern of SFT-T1 and SFT-T2 cells. (**a**) SFT typical mesenchymal tumor cell shape as compared to HCC827 and Met5a. (**b**) Protein expression concerning vimentin and bcl-2 in SFT-T1 and SFT-T2 cells as compared to Met5a and HCC827 cells used as negative and positive controls, respectively. ß-actin served as loading control. (**c**) Verification of NAB2-STAT6 expression by RT-PCR. GAPDH served as loading control. (**d**) Expression of wild-type and NAB2-STAT6 fusion protein in SFT and control cells as indicated.

**Figure 4 cancers-14-05602-f004:**
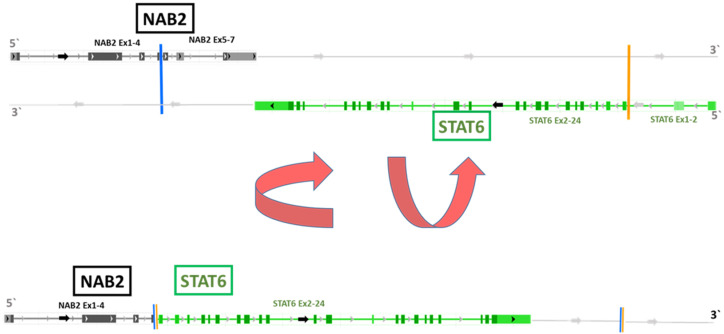
NAB2-STAT6 fusion variant as detected in all four SFT cell models by NGS. The lower part of the figure depicts the NAB2-STAT6 fusion variant (NAB2 exon 4, Stat6 exon 2) obtained by inversion of individual NAB2 and STAT6 strands shown in the upper part. The fusion gene was present in both cell lines (SFT-T1 and SFT-T2) and in all five surgically removed tumor samples, as found by NGS and PCR investigations. This fusion variant is associated with pleural SFT as published [26].

**Figure 5 cancers-14-05602-f005:**
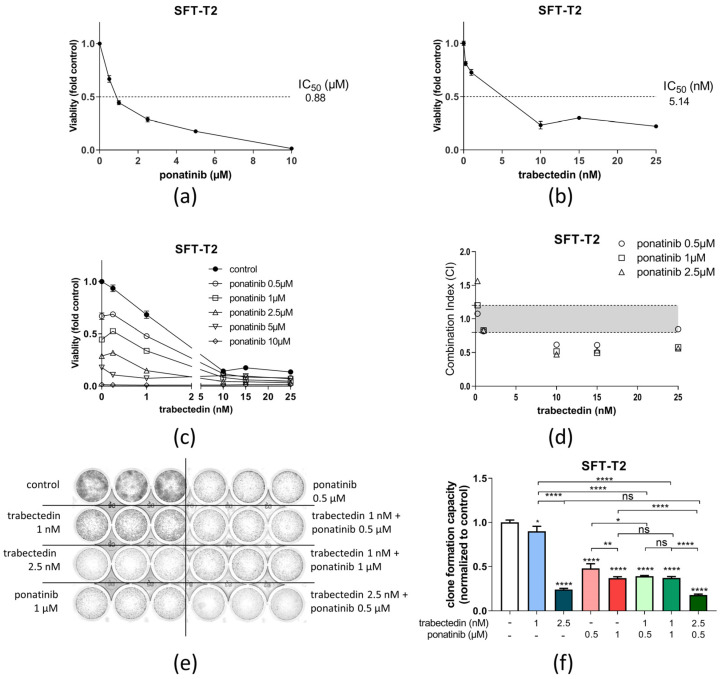
Ponatinib and trabectedin are active against SFT-T2 cell growth and synergize in vitro. Growth inhibitory effects of (**a**) ponatinib or (**b**) trabectedin as single agents in SFT-T2 cells assessed via MTT assay for 72 h of treatment. (**c**) Combination of ponatinib with trabectedin shows synergistic anticancer activity. (**d**) Corresponding CI values. CI < 0.9, synergism; CI = 0.9–1.2, additive effects; or CI > 1.2, antagonism. (**e**) Effects of ponatinib as well as trabectedin as single agents and in combination on SFT clone formation capacity over a treatment period of nine days. (**f**) Quantification of the clone formation assay. * *p* < 0.05, ** *p* < 0.005 and **** *p* < 0.0001.

**Figure 6 cancers-14-05602-f006:**
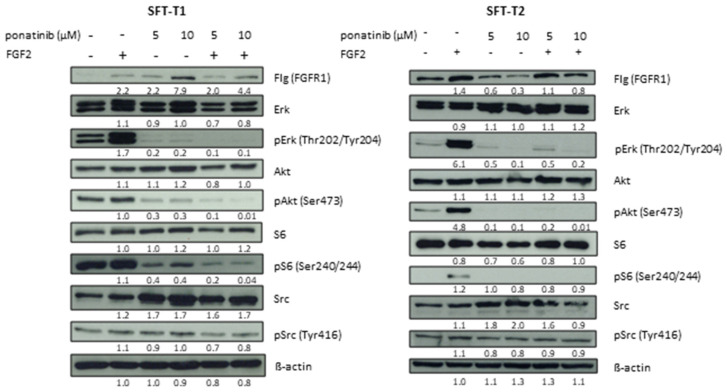
Ponatinib inhibits MAPK and PI3K/Akt signaling cascades in SFT-T1 and -T2 cells. Expression and phosphorylation levels of respective proteins after 1 h of treatment with indicated concentrations of ponatinib with or without stimulation with FGF2 for 15 min were analyzed by Western blotting. ß-actin served as loading control. Numbers below represent quantified signal intensities normalized to respective ß-actin relative to untreated control.

**Figure 7 cancers-14-05602-f007:**
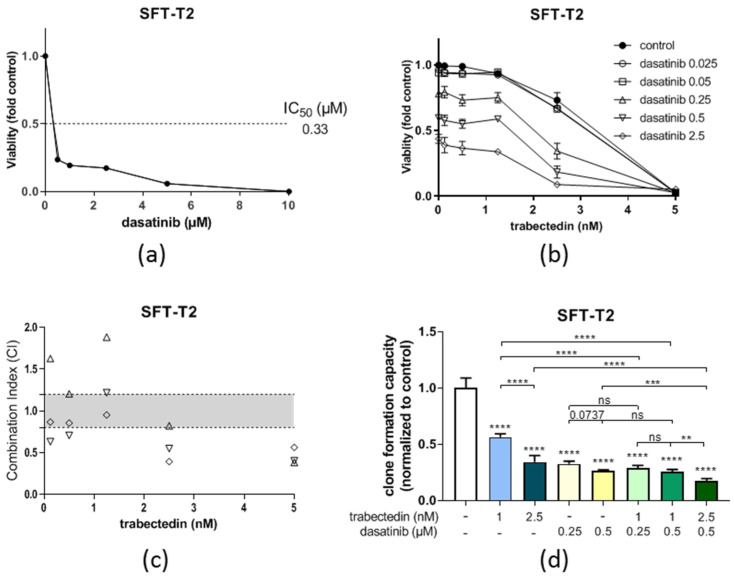
Dasatinib is active against SFT-T2 growth in vitro and enhances the anticancer activity of trabectedin. (**a**) Growth inhibitory effects of dasatinib as single agents in SFT-T2 cells assessed via MTT assay for 72 h of treatment. (**b**) Combination of dasatinib with trabectedin shows additive or even synergistic anticancer activity. (**c**) Corresponding CI values. CI < 0.9, synergism; CI = 0.9–1.2, additive effects; or CI > 1.2, antagonism. (**d**) Quantification of the effects of dasatinib as well as trabectedin as single agents and in combination on SFT-T2 clone formation capacity over a treatment period of nine days. ** *p* < 0.005, *** *p* < 0.0005, and **** *p* < 0.0001.

**Table 1 cancers-14-05602-t001:** Clinical, pathological, and genetic characterization of the study population and the respective cell lines.

SFT#	Age	Sex	Risk Group	CD34	bcl-2	CD99	STAT6	Ki67	*NAB2-STAT6*	*NAB2-STAT6*
				Tumor						Cell Line
T92	72	male	intermediate	+	+	+	+	15	+	−
T1	50	male	low	+	+	+	+	1	+	+
T2	45	female	intermediate	+	+	+	+	5	+	+
T8	34	female	intermediate	+	+	+	+	20	+	−
T9	76	female	high	+	+	+	+	25	+	−

**Abbreviations:** +—positive; −—negative; Ki67—% of positive tumor cells; risk group according to Demicco et al. [25].

**Table 2 cancers-14-05602-t002:** In vitro drug responsiveness of two SFT cell models including respective main targets of the drugs.

	IC_50_ ± SD		
Drug	SFT-T1	SFT-T2	Main Target(s)
Cisplatin (µM)	16.38 ± 1.91	14.64 ± 4.91	DNA synthesis
Paclitaxel (µM)	0.032 ± 0.04	0.451 ± 0.57	cytoskeletal components
Vincristine (nM)	37.50 ± 9.09	36.54 ± 10.08	microtubule formation
Venetoclax (µM)	27.91 ± 5.34	18.03 ± 4.64	bcl-2
Obatoclax (nM)	258.55 ± 84.24	237.69 ± 33.68	bcl-2
Doxorubicin (nM)	555.36 ± 327.33	410.51 ± 193.32	DNA topoisomerase II
Imatinib (µM)	27.28 ± 6.92	18.18 ± 2.70	v-Abl, c-Kit, and PDGFR
Stattic (µM)	3.04 ± 1.09	5.30 ± 1.52	STAT3
Trabectedin (nM)	4.36 ± 0.73	3.54 ± 0.98	DNA damaging, blocks DNA binding of FUS-CHOP
Ponatinib (µM)	0.47 ± 0.07	1.89 ± 0.98	FGFR, Abl, PDGFRα, VEGFR2, and Src
Nintedanib (µM)	3.59 ± 0.89	3.63 ± 1.84	VEGFR1-3, FGFR1-3, PDGFR α, and β
Dasatinib (µM)	0.39 ± 0.02	0.40 ± 0.19	Abl, Src, and c-Kit
PD173074 (µM)	2.32 ± 0.73	2.46 ± 0.46	FGFR1 and VEGFR2

**Abbreviations:** IC_50_—half maximal inhibitory concentration; SD—standard deviation.

## Data Availability

Upon reasonable request, all data and materials are available from the first author.

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
