# Peer review of "Trabectedin Is Active against Two Novel, Patient-Derived Solitary Fibrous Pleural Tumor Cell Lines and Synergizes with Ponatinib"

_cancers, 2022, doi:10.3390/cancers14225602_

Round 1

Reviewer 1 Report

This is a novel an interesting study on SFT, providing valuable insights into therapeutic approaches with available drugs. Of course, we do not know if the suggested combination will also work in patients as well, but the study is well done and the results are sound. 

Author Response

Reviewer 1: This is a novel an interesting study on SFT, providing valuable insights into therapeutic approaches with available drugs. Of course, we do not know if the suggested combination will also work in patients as well, but the study is well done and the results are sound. 

Authors Reply: The authors want to thank the referee for the positive comment and his efforts. We hope that our preclinical study can serve as basis for future clinical trials to test our results also in the clinical setting.

Reviewer 2 Report

The manuscript entitled "Trabectedin is Active Against Two Novel Patient-derived Solitary Fibrous Pleura Tumor Cell Lines, and Synergizes with Ponatinib" by Ghanim et al. characterizes SFT diseases related to patient cell lines, having a specific NAB2-STAT6 gene fusion. The authors have further explored the effectiveness of drugs on these cell lines to find an effective treatment.  

This is a well-written manuscript to be accepted with minor revisions.

·      The authors need to provide better wester blotting gels, for example,

o   fig 3b, middle panel; fig 3b, for the bottom image, the loading control amount is different.

·      “Tumor cells were positive with regard to membranous and cytoplasmic expression of CD34, bcl-2 and CD99 and furthermore showed nuclear STAT6 positivity (Figure 2b, c, d, and f, respectively).” The authors need to show images for negative control.

·      “Both cell lines demonstrated stable in vitro proliferation and a mesenchymal growth pattern as shown in Figure 3a” Again, the authors need to show data for control cell lines. Authors also need to show time-dependent cell proliferation.

·      “the presence of the fusion oncogene detected by NGS was verified by reverse PCR (Figure 3c).” The authors also need to perform DNA/RNA sequencing to verify the breakpoint of fusion between two genes.

·      For figure 3c, the authors need to mark different lanes.

·      For table 2, the authors need to show the data as a supporting figure from where they have determined the IC50 values.

·      There is no error bar in fig 5a, 7a.

·      Do a quantification for the fig 6.

·      Check for typo. For example,

o   with an annual incidence of less than 0.1 per 100.000 in Europe

Author Response

Reviewer 2: The manuscript entitled "Trabectedin is Active Against Two Novel Patient-derived Solitary Fibrous Pleura Tumor Cell Lines, and Synergizes with Ponatinib" by Ghanim et al. characterizes SFT diseases related to patient cell lines, having a specific NAB2-STAT6 gene fusion. The authors have further explored the effectiveness of drugs on these cell lines to find an effective treatment.  

This is a well-written manuscript to be accepted with minor revisions.

The authors need to provide better wester blotting gels, for example,

o   fig 3b, middle panel; fig 3b, for the bottom image, the loading control amount is different. Dina

Authors Reply: Thank you for your time and effort to review our manuscript and for your suggestions for improvement. The Vimentin and bcl-2 blot (fig 3b) have been revised according to the referee´s suggestions.

 “Tumor cells were positive with regard to membranous and cytoplasmic expression of CD34, bcl-2 and CD99 and furthermore showed nuclear STAT6 positivity (Figure 2b, c, d, and f, respectively).” The authors need to show images for negative control.

Authors Reply: Thank you for this excellent comment. Negative isotype controls can now be found in supplementary figure S1. Fibromatosis is now added to figure 2 as pathological counterpart of SFT.

“Both cell lines demonstrated stable in vitro proliferation and a mesenchymal growth pattern as shown in Figure 3a” Again, the authors need to show data for control cell lines. Authors also need to show time-dependent cell proliferation.

Authors Reply: 

Thank you for this comment. Images presenting epithelial cell morphology of control cell lines HCC827 and Met5a have been added to Figure 3a. Regarding the second part of the question, time-dependent cell proliferation was measured after 24, 48, and 72 h and is now given in supplementary figure S2

“the presence of the fusion oncogene detected by NGS was verified by reverse PCR (Figure 3c).” The authors also need to perform DNA/RNA sequencing to verify the breakpoint of fusion between two genes.

Authors Reply: Thank you for this valuable comment! We now express clearer in the text that the fusion was characterized on the DNA level by NGS and afterwards confirmed on the RNA level by RT-PCR. The corresponding sentence on page 5 has been adjusted to “In addition, the presence of the fusion oncogene detected by NGS was verified on the RNA level by reverse transcription PCR (Figure 3c).”

For figure 3c, the authors need to mark different lanes.

Authors Reply: Thank you for your comment. The lanes of SFT-T1 and SFT-T2 in Figure 3c have been marked accordingly.

For table 2, the authors need to show the data as a supporting figure from where they have determined the IC50 values.

Authors Reply: Following the referee’s suggestion, we included representative dose-response curves as additional Figure S3 in the Supplementary Material.

There is no error bar in fig 5a, 7a.

Authors Reply: Thank you for your observation. We have now modified both figures according to the referees suggestion.

Please note, in Figure 7a error bars are too small to be seen underneath the data points. For clarification we provide a graph with smaller symbols for the reviewer.

Do a quantification for the fig 6.

Authors Reply: Thank you for your suggestion, we have now adapted fig 6 according to your comment and included the quantified expression levels beneath each Western blot signal.

Check for typo. For example, with an annual incidence of less than 0.1 per 100.000 in Europe

Authors Reply: Thank you, we have checked again for typos and corrected the respective errors. Furthermore, some repetitions in the methods section have been erased according to the editorial team’s suggestion

Reviewer 3 Report

Very interesting study that presents two novel characterized SFT cell lines that were tested for several single drugs and combinations. 

These are my remarks / suggestions:

1. I wonder if the addition of human EGF to the growth medium of the cells could have an influence on your results since ponatinib is described to inhibit STAT3 activity driven by EGF/EGFR.

2. In fig 1 I would change the color of the arrows above the time line. Now it seems that these are also unstable cell lines (according to the color)

3. In the legend of fig 3 you state that Met5a and HCC827 cells are used as positive and negative controls, respectively. By looking at the blot I would say that it is the other way around (Met5a seems rather negative).

4. Line 282: you state that nintedanib also showed pronounced activity, but if that's the case than why not PD173074 which shows even a lower IC50?

5. Line 283: Obatoclax was stated to be moderately active, although it shows a lower IC50 compared to ponatinib and dasatinib.

6. I wonder why you don't show the curves (fig5 a-c) for SFT-T1.

7. Why did you use different doses of trabectedin in the combination experiment for T1 and T2 while the IC50 value did not vary that much.

8. Line 327: you state that the results for T1 are comparable, however you see some differences (higher FGFR1 after ponatinib treatment, pS6 is present at baseline in T1 but not in T2). It would be good to explain those and show them both in the article. 

9. In Fig 6 and S2 you see an upregulation of Src after ponatinib treatment. Please explain.

10. For the combination of trabectidin and dasatinib I miss the quantification of the combination effects in T1 (only shown in fig 7d for T2).

Author Response

Reviewer 3: Very interesting study that presents two novel characterized SFT cell lines that were tested for several single drugs and combinations. 

These are my remarks / suggestions:

  1. I wonder if the addition of human EGF to the growth medium of the cells could have an influence on your results since ponatinib is described to inhibit STAT3 activity driven by EGF/EGFR.

Authors Reply: Thank you for taking time and reviewing our paper. We have now also checked for this important issue.

Following your suggestion, we tested expression levels of EGFR and STAT3 as well as their phosphorylation with or without ponatinib treatment (graph is attached). Addition of ponatinib induced enhanced expression of EGFR and STAT3 together with an increased phosphorylation of EGFR rather than the suggested potential downregulation. Additionally, no considerable regulation of STAT3 phosphorylation was observed under ponatinib treatment. Hence, inhibition of EGFR and STAT3 activation via ponatinib treatment seems to be of less importance in our SFTP cell models.

  1. In fig 1 I would change the color of the arrows above the time line. Now it seems that these are also unstable cell lines (according to the color)

Authors Reply: Thank you for your suggestion; we have now adapted Figure 1 according to the referee´s comment.

  1. In the legend of fig 3 you state that Met5a and HCC827 cells are used as positive and negative controls, respectively. By looking at the blot I would say that it is the other way around (Met5a seems rather negative).

Authors Reply: The authors want to thank the referee for carefully reviewing our manuscript. The figure legend is now corrected according to the referees´ suggestion

  1. Line 282: you state that nintedanib also showed pronounced activity, but if that's the case than why not PD173074 which shows even a lower IC50?

Authors Reply:  Thank you for this comment. We have now changed the respective part of our results section and also underlined the role of FGFR targeting drugs one more time in the discussion section (page 15).

  1. Line 283: Obatoclax was stated to be moderately active, although it shows a lower IC50 compared to ponatinib and dasatinib.

Authors Reply: Thank you for this comment. We have now stated that the activity of obatoclax was moderate when compared to colorectal and bladder cancer cell lines (page 11). Please also compare the corresponding literature from Or et al [1] regarding CRC and Steele et al [2] regarding bladder cancer. Both citations are now also included in the manuscript.

  1. I wonder why you don't show the curves (fig5 a-c) for SFT-T1.

Authors Reply: Thank you for your suggestion. The respective results for the combination experiment and the CI value determination are now also added to the supplementary material of our manuscript in Figure S4.

  1. Why did you use different doses of trabectedin in the combination experiment for T1 and T2 while the IC50 value did not vary that much.

Authors Reply: Thank you for this comment, we now used the same doses in both cell lines as suggested by the referee

  1. Line 327: you state that the results for T1 are comparable, however you see some differences (higher FGFR1 after ponatinib treatment, pS6 is present at baseline in T1 but not in T2). It would be good to explain those and show them both in the article. 

Authors Reply: Thank you for your comment. Both figures are now presented side by side in the main manuscript and the respective differences are now explained in the results section according to the referees’ suggestion on page 12.

  1. In Fig 6 and S2 you see an upregulation of Src after ponatinib treatment. Please explain.

Authors Reply: Thank you for this excellent comment. Indeed, Src expression but not phosphorylation increased after ponatinib exposure. This finding is in line with literature, where ponatinib is a well-known (among other tyrosine kinases) Src inhibitor. However, as explained by Zeng et al, a dose- or specificity-dependent impact on the Src family kinase might be observed after ponatinib exposure. This activating effect on Src is also linked to cardiovascular and thrombotic events after Abl-targeting therapy (Compare Zeng et al [3]).

  1. For the combination of trabectedin and dasatinib I miss the quantification of the combination effects in T1 (only shown in fig 7d for T2).

Authors Reply: Thank you; we have adapted Figure S5 according to the referee’s suggestion

  1. Or, C.R.; Chang, Y.; Lin, W.C.; Lee, W.C.; Su, H.L.; Cheung, M.W.; Huang, C.P.; Ho, C.; Chang, C.C. Obatoclax, a Pan-BCL-2 Inhibitor, Targets Cyclin D1 for Degradation to Induce Antiproliferation in Human Colorectal Carcinoma Cells. Int J Mol Sci 2016, 18, doi:10.3390/ijms18010044.
  2. Steele, T.M.; Talbott, G.C.; Sam, A.; Tepper, C.G.; Ghosh, P.M.; Vinall, R.L. Obatoclax, a BH3 Mimetic, Enhances Cisplatin-Induced Apoptosis and Decreases the Clonogenicity of Muscle Invasive Bladder Cancer Cells via Mechanisms That Involve the Inhibition of Pro-Survival Molecules as Well as Cell Cycle Regulators. Int J Mol Sci 2019, 20, doi:10.3390/ijms20061285.
  3. Zeng, P.; Schmaier, A. Ponatinib and other CML Tyrosine Kinase Inhibitors in Thrombosis. Int J Mol Sci 2020, 21, doi:10.3390/ijms21186556.

Round 2

Reviewer 2 Report

The authors have addressed most of the concerns. This article can be published now.